# Weight Agnostic Neural Networks

**Adam Gaier**[†]
Bonn-Rhein-Sieg University of Applied Sciences
Inria / CNRS / Université de Lorraine
adam.gaier@h-brs.de

**David Ha**
Google Brain
Tokyo, Japan
hadavid@google.com

## Abstract

Not all neural network architectures are created equal, some perform much better than others for certain tasks. But how important are the weight parameters of a neural network compared to its architecture? In this work, we question to what extent neural network architectures alone, without learning any weight parameters, can encode solutions for a given task. We propose a search method for neural network architectures that can already perform a task without any explicit weight training. To evaluate these networks, we populate the connections with a single shared weight parameter sampled from a uniform random distribution, and measure the expected performance. We demonstrate that our method can find minimal neural network architectures that can perform several reinforcement learning tasks without weight training. On a supervised learning domain, we find network architectures that achieve much higher than chance accuracy on MNIST using random weights. Interactive version of this paper at https://weightagnostic.github.io/

## 1 Introduction

In biology, precocial species are those whose young already possess certain abilities from the moment of birth. There is evidence to show that lizard [75] and snake [13, 78] hatchlings already possess behaviors to escape from predators. Shortly after hatching, ducks are able to swim and eat on their own [107], and turkeys can visually recognize predators [27]. In contrast, when we train artificial agents to perform a task, we typically choose a neural network architecture we believe to be suitable for encoding a policy for the task, and find the weight parameters of this policy using a learning algorithm. Inspired by precocial behaviors evolved in nature, in this work, we develop neural networks with architectures that are naturally capable of performing a given task even when their weight parameters are randomly sampled. By using such neural network architectures, our agents can already perform well in their environment without the need to learn weight parameters.

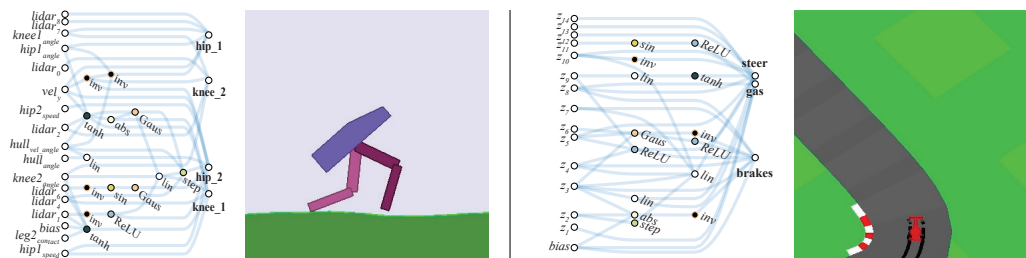

Figure 1: *Examples of Weight Agnostic Neural Networks: Bipedal Walker (left), Car Racing (right)* We search for architectures by deemphasizing weights. In place of training, networks are assigned a single shared weight value at each rollout. Architectures that are optimized for expected performance over a wide range of weight values are still able to perform various tasks without weight training.

---

[†]Work done while at Google Brain.

Decades of neural network research have provided building blocks with strong inductive biases for various task domains. Convolutional networks [23, 57] are especially suited for image processing [16]. Recent work [41, 113] demonstrated that even randomly-initialized CNNs can be used effectively for image processing tasks such as superresolution, inpainting and style transfer. Schmidhuber et al. [99] have shown that a randomly-initialized LSTM [45] with a learned linear output layer can predict time series where traditional reservoir-based RNNs [47, 95] fail. More recent developments in self-attention [116] and capsule [96] networks expand the toolkit of building blocks for creating architectures with strong inductive biases for various tasks. Fascinated by the intrinsic capabilities of randomly-initialized CNNs and LSTMs, we aim to search for *weight agnostic neural networks*, architectures with strong inductive biases that can already perform various tasks with random weights.

In order to find neural network architectures with strong inductive biases, we propose to search for architectures by deemphasizing the importance of weights. This is accomplished by (**1**) assigning a single shared weight parameter to every network connection and (**2**) evaluating the network on a wide range of this single weight parameter. In place of optimizing weights of a fixed network, we optimize instead for architectures that perform well over a wide range of weights. We demonstrate our approach can produce networks that can be expected to perform various continuous control tasks with a random weight parameter. As a proof of concept, we also apply our search method on a supervised learning domain, and find it can discover networks that, even without explicit weight training, can achieve a much higher than chance test accuracy of $\sim 92\%$ on MNIST. We hope our demonstration of such weight agnostic neural networks will encourage further research exploring novel neural network building blocks that not only possess useful inductive biases, but can also learn using algorithms that are not necessarily limited to gradient-based methods.[2]

## 2 Related Work

Our work has connections to existing work not only in deep learning, but also to various other fields:

**Architecture Search**  Search algorithms for neural network topologies originated from the field of evolutionary computing [2, 8, 17, 24, 31, 39, 54, 60, 72, 73, 83, 88, 112, 119, 121]. Our method is based on NEAT [106], an established topology search algorithm notable for its ability to optimize the weights and structure of networks simultaneously. In order to achieve state-of-the-art results, recent methods narrow the search space to architectures composed of basic building blocks with strong domain priors such as CNNs [66, 74, 91, 123], recurrent cells [49, 74, 123] and self-attention [103]. It has been shown that random search can already achieve SOTA results if such priors are used [65, 90, 100]. The inner loop for training the weights of each candidate architecture before evaluation makes the search costly, although efforts have been made to improve efficiency [9, 67, 87]. In our approach, we evaluate architectures without weight training, bypassing the costly inner loop, similar to the random trial approach in [44, 102] that evolved architectures to be more weight tolerant.

**Bayesian Neural Networks**  The weight parameters of a BNN [4, 5, 25, 43, 70, 80] are not fixed values, but sampled from a distribution. While the parameters of this distribution can be learned [28, 37, 38, 55], the number of parameters is often greater than the number of weights. Recently, Neklyudov et al. [81] proposed variance networks, which sample each weight from a distribution with a zero mean and a learned variance parameter, and show that ensemble evaluations can improve performance on image recognition tasks. We employ a similar approach, sampling weights from a fixed uniform distribution with zero mean, as well as evaluating performance on network ensembles.

**Algorithmic Information Theory**  In AIT [104], the Kolmogorov complexity [52] of a computable object is the minimum length of the program that can compute it. The Minimal Description Length (MDL) [32, 93, 94] is a formalization of Occam's razor, in which a good model is one that is best at compressing its data, including the cost of describing of the model itself. Ideas related to MDL for making neural networks "simple" was proposed in the 1990s, such as simplifying networks by soft-weight sharing [82], reducing the amount of information in weights by making them noisy [43], and simplifying the search space of its weights [98]. Recent works offer a modern treatment [7] and application [63, 111] of these principles in the context of larger, deep neural network architectures.

While the aforementioned works focus on the information capacity required to represent the *weights* of a predefined network architecture, in this work we focus on finding minimal *architectures* that can

represent solutions to various tasks. As our networks still require weights, we borrow ideas from AIT and BNN, and take them a bit further. Motivated by MDL, in our approach, we apply weight-sharing to the entire network and treat the weight as a random variable sampled from a fixed distribution.

**Network Pruning** By removing connections with small weight values from a trained neural network, pruning approaches [33, 36, 40, 59, 64, 68, 69, 71, 77] can produce sparse networks that keep only a small fraction of the connections, while maintaining similar performance on image classification tasks compared to the full network. By retaining the original weight initialization values, these sparse networks can even be trained from scratch to achieve a higher test accuracy [22, 61] than the original network. Similar to our work, a concurrent work [122] found pruned networks that can achieve image classification accuracies that are much better than chance even with randomly initialized weights.

Network pruning is a complementary approach to ours; it starts with a full, trained network, and takes away connections, while in our approach, we start with no connections, and add complexity as needed. Compared to our approach, pruning requires prior training of the full network to obtain useful information about each weight in advance. In addition, the architectures produced by pruning are limited by the full network, while in our method there is no upper bound on the network's complexity.

**Neuroscience** A *connectome* [101] is the "wiring diagram" or mapping of all neural connections of the brain. While it is a challenge to map out the human connectome [105], with our 90 billion neurons and 150 trillion synapses, the connectome of simple organisms such as roundworms [115, 117] has been constructed, and recent works [20, 108] mapped out the entire brain of a small fruit fly. A motivation for examining the connectome, even of an insect, is that it will help guide future research on how the brain learns and represents memories in its connections. For humans it is evident, especially during early childhood [46, 110], that we learn skills and form memories by forming new synaptic connections, and our brain rewires itself based on our new experiences [6, 11, 18, 51].

The connectome can be viewed as a graph [12, 42, 114], and analyzed using rich tools from graph theory, network science and computer simulation. Our work also aims to learn network graphs that can encode skills and knowledge for an artificial agent in a simulation environment. By deemphasizing learning of weight parameters, we encourage the agent instead to develop ever-growing networks that can encode acquired skills based on its interactions with the environment. Like the connectome of simple organisms, the networks discovered by our approach are small enough to be analyzed.

## 3   Weight Agnostic Neural Network Search

Creating network architectures which encode solutions is a fundamentally different problem than that addressed by neural architecture search (NAS). The goal of NAS techniques is to produce architectures which, once trained, outperform those designed by humans. It is never claimed that the solution is innate to the structure of the network. Networks created by NAS are exceedingly 'trainable' – but no one supposes these networks will solve the task without training the weights. The weights *are* the solution; the found architectures merely a better substrate for the weights to inhabit.

To produce architectures that themselves encode solutions, the importance of weights must be minimized. Rather than judging networks by their performance with optimal weight values, we can instead measure their performance when their weight values are drawn from a random distribution. Replacing weight training with weight sampling ensures that performance is a product of the network topology alone. Unfortunately, due to the high dimensionality, reliable sampling of the weight space is infeasible for all but the simplest of networks.

Though the curse of dimensionality prevents us from efficiently sampling high dimensional weight spaces, by enforcing weight-sharing on *all* weights, the number of weight values is reduced to one. Systematically sampling a single weight value is straight-forward and efficient, enabling us to approximate network performance in only a handful of trials. This approximation can then be used to drive the search for ever better architectures.

The search for these weight agnostic neural networks (WANNs) can be summarized as follows (See Figure 2 for an overview): **(1)** An initial population of minimal neural network topologies is created, **(2)** each network is evaluated over multiple rollouts, with a different shared weight value assigned at each rollout, **(3)** networks are ranked according to their performance *and* complexity, and **(4)** a new population is created by varying the highest ranked network topologies, chosen probabilistically through tournament selection [76]. The algorithm then repeats from **(2)**, yielding weight agnostic topologies of gradually increasing complexity that perform better over successive generations.

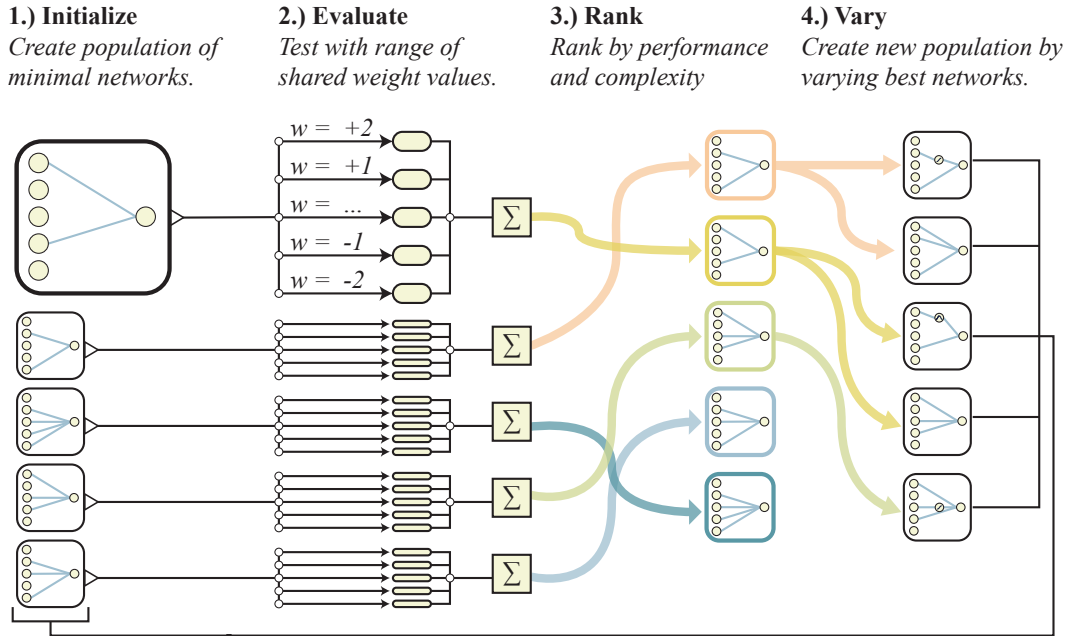

**1.) Initialize**
*Create population of minimal networks.*

**2.) Evaluate**
*Test with range of shared weight values.*

**3.) Rank**
*Rank by performance and complexity*

**4.) Vary**
*Create new population by varying best networks.*

Figure 2: *Overview of Weight Agnostic Neural Network Search*
Weight Agnostic Neural Network Search avoids weight training while exploring the space of neural network topologies by sampling a single shared weight at each rollout. Networks are evaluated over several rollouts. At each rollout a value for the single shared weight is assigned and the cumulative reward over the trial is recorded. The population of networks is then ranked according to their performance and complexity. The highest ranking networks are then chosen probabilistically and varied randomly to form a new population, and the process repeats.

**Topology Search** The operators used to search for neural network topologies are inspired by the well-established neuroevolution algorithm NEAT [106]. While in NEAT the topology and weight values are optimized simultaneously, we ignore the weights and apply only topological search operators.

The initial population is composed of sparsely connected networks, networks with no hidden nodes and only a fraction of the possible connections between input and output. New networks are created by modifying existing networks using one of three operators: insert node, add connection, or change activation (Figure 3). To insert a node, we split an existing connection into two connections that pass through this new hidden node. The activation function of this new node is randomly assigned. New connections are added between previously unconnected nodes, respecting the feed-forward property of the network. When activation functions of hidden nodes are changed, they are assigned at random. Activation functions include both the common (e.g. linear, sigmoid, ReLU) and more exotic (Gaussian, sinusoid, step), encoding a variety of relationships between inputs and outputs.

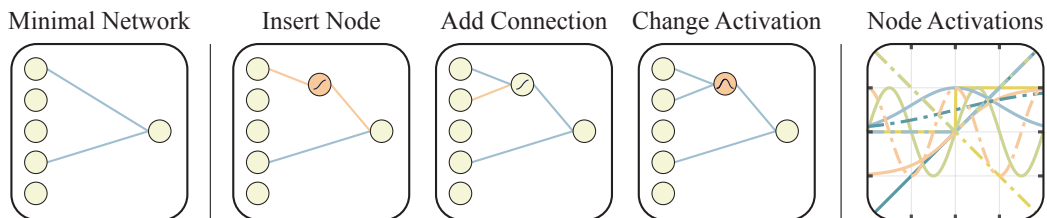

Minimal Network    Insert Node    Add Connection    Change Activation    Node Activations

Figure 3: *Operators for Searching the Space of Network Topologies*
*Left:* A minimal network topology, with input and outputs only partially connected.
*Middle:* Networks are altered in one of three ways. *Insert Node*: a new node is inserted by splitting an existing connection. *Add Connection*: a new connection is added by connecting two previously unconnected nodes. *Change Activation*: the activation function of a hidden node is reassigned.
*Right:* Possible activation functions (linear, step, sin, cosine, Gaussian, tanh, sigmoid, absolute value, invert (i.e. negative linear), ReLU) shown over the range $[2, 2]$.

**Performance and Complexity** Network topologies are evaluated using several shared weight values. At each rollout a new weight value is assigned to *all* connections, and the network is tested on the task. In these experiments we used a fixed series of weight values ($[-2, -1, -0.5, +0.5, +1, +2]$) to decrease the variance between evaluations.[3] We calculate the mean performance of a network topology by averaging its cumulative reward over all rollouts using these different weight values.

Motivated by algorithmic information theory [104], we are not interested in searching merely for *any* weight agnostic neural networks, but networks that can be described with a minimal description length [32, 93, 94]. Given two different networks with similar performance we prefer the simpler network. By formulating the search as a multi-objective optimization problem [53, 79] we take into account the size of the network as well as its performance when ranking it in the population.

We apply the connection cost technique from [15] shown to produce networks that are more simple, modular, and evolvable. Networks topologies are judged based on three criteria: mean performance over all weight values, max performance of the single best weight value, and the number of connections in the network. Rather than attempting to balance these criteria with a hand-crafted reward function for each new task, we rank the solutions based on dominance relations [19].

Ranking networks in this way requires that any increase in complexity is accompanied by an increase in performance. While encouraging minimal and modular networks, this constraint can make larger structural changes – which may require several additions before paying off – difficult to achieve. To relax this constraint we rank by complexity only probabilistically: in 80% of cases networks are ranked according to mean performance and the number of connections, in the other 20% ranking is done by mean performance and max performance.

## 4 Experimental Results

**Continuous Control** Weight agnostic neural networks (WANNs) are evaluated on three continuous control tasks. The first, `CartPoleSwingUp`, is a classic control problem where, given a cart-pole system, a pole must be swung from a resting to upright position and then balanced, without the cart going beyond the bounds of the track. The swingup task is more challenging than the simpler `CartPole` [10], where the pole starts upright. Unlike the simpler task, it cannot be solved with a linear controller [89, 109]. The reward at every timestep is based on the distance of the cart from track edge and the angle of the pole. Our environment is closely based on the one described in [26, 124].

The second task, `BipedalWalker-v2` [10], is to guide a two-legged agent across randomly generated terrain. Rewards are awarded for distance traveled, with a cost for motor torque to encourage efficient movement. Each leg is controlled by a hip and knee joint in reaction to 24 inputs, including LIDAR sensors which detect the terrain and proprioceptive information such as the agent's joint speeds. Compared to the low dimensional `CartPoleSwingUp`, `BipedalWalker-v2` has a non-trivial number of possible connections, requiring WANNs to be selective about the wiring of inputs to outputs.

The third, `CarRacing-v0` [10], is a top-down car racing from pixels environment. A car, controlled with three continuous commands (gas, steer, brake) is tasked with visiting as many tiles as possible of a randomly generated track within a time limit. Following the approach described in [35], we delegate the pixel interpretation element of the task to a pre-trained variational autoencoder [50, 92] (VAE) which compresses the pixel representation to 16 latent dimensions. These dimensions are given as input to the network. The use of learned features tests the ability of WANNs to learn abstract associations rather than encoding explicit geometric relationships between inputs.

Hand-designed networks found in the literature [34, 35] are compared to the best weight agnostic networks found for each task. We compare the mean performance over 100 trials under 4 conditions:

1. *Random weights*: individual weights drawn from $\mathcal{U}(-2, 2)$;
2. *Random shared weight*: a single shared weight drawn from $\mathcal{U}(-2, 2)$;
3. *Tuned shared weight*: the highest performing shared weight value in range $(-2, 2)$;
4. *Tuned weights*: individual weights tuned using population-based REINFORCE [118].

Table 1: *Performance of Randomly Sampled and Trained Weights for Continuous Control Tasks*
We compare the cumulative reward (average of 100 random trials) of the best weight agnostic network architectures found with standard feed forward network policies commonly used in previous work (i.e. [34, 35]). The intrinsic bias of a network topology can be observed by measuring its performance using a shared weight sampled from a uniform distribution. By tuning this shared weight parameter we can measure its maximum performance. To facilitate comparison to baseline architectures we also conduct experiments where networks are allowed unique weight parameters and tuned.

| **Swing Up** | Random Weights | Random Shared Weight | Tuned Shared Weight | Tuned Weights |
|---|---|---|---|---|
| WANN | **57 ± 121** | **515 ± 58** | **723 ± 16** | **932 ± 6** |
| Fixed Topology | 21 ± 43 | 7 ± 2 | 8 ± 1 | 918 ± 7 |

| **Biped** | Random Weights | Random Shared Weight | Tuned Shared Weight | Tuned Weights |
|---|---|---|---|---|
| WANN | **-46 ± 54** | **51 ± 108** | **261 ± 58** | 332 ± 1 |
| Fixed Topology | -129 ± 28 | -107 ± 12 | -35 ± 23 | **347 ± 1** [34] |

| **CarRacing** | Random Weights | Random Shared Weight | Tuned Shared Weight | Tuned Weights |
|---|---|---|---|---|
| WANN | **-69 ± 31** | **375 ± 177** | **608 ± 161** | 893 ± 74 |
| Fixed Topology | -82 ± 13 | -85 ± 27 | -37 ± 36 | **906 ± 21** [35] |

The results are summarized in Table 1.[4] In contrast to the conventional fixed topology networks used as baselines, which only produce useful behaviors after extensive tuning, WANNs perform even with random shared weights. Though their architectures encode a strong bias toward solutions, WANNs are not completely independent of the weight values – they do fail when individual weight values are assigned randomly. WANNs function by encoding relationships between inputs and outputs, and so while the importance of the magnitude of the weights is not critical, their consistency, especially consistency of sign, is. An added benefit of a single shared weight is that it becomes trivial to tune this single parameter, without requiring the use of gradient-based methods.

The best performing shared weight value produces satisfactory if not optimal behaviors: a balanced pole after a few swings, effective if inefficient gaits, wild driving behaviour that cuts corners. These basic behaviors are encoded entirely within the architecture of the network. And while WANNs are able to perform without training, this predisposition does not prevent them from reaching similar state-of-the-art performance when the weights *are* trained.

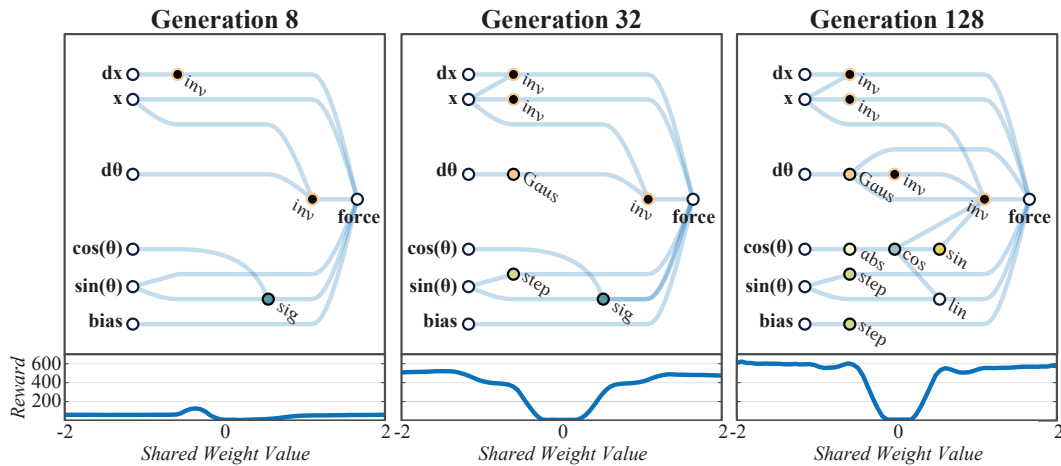

Figure 4: *Development of Weight Agnostic Neural Network Topologies Over Time*
*Generation 8:* An early network which performs poorly with nearly all weights.
*Generation 32:* Relationships between the position of the cart and velocity of the pole are established. The tension between these relationships produces both centering and swing-up behavior.
*Generation 128:* Complexity is added to refine the balancing behavior of the elevated pole.

As the networks discovered are small enough to interpret, we can derive insights into how they function by looking at network diagrams (See Figure 4). Examining the development of a WANN which solves `CartPoleSwingUp` is also illustrative of how relationships are encoded within an

architecture. In the earliest generations the space of networks is explored in an essentially random fashion. By generation 32, preliminary structures arise which allow for consistent performance: the three inverters applied to the $x$ position keep the cart from leaving the track. The center of the track is at 0, left is negative, right is positive. By applying positive force when the cart is in a negative position and vice versa a strong attractor towards the center of the track is encoded.

The interaction between the regulation of position and the Gaussian activation on $d\theta$ is responsible for the swing-up behavior, also developed by generation 32. At the start of the trial the pole is stationary: the Gaussian activation of $d\theta$ is 1 and force is applied. As the pole moves toward the edge the nodes connected to the $x$ input, which keep the cart in the center, begin sending an opposing force signal. The cart's progress toward the edge is slowed and the change in acceleration causes the pole to swing, increasing $d\theta$ and so decreasing the signal that is pushing the cart toward the edge. This slow down causes further acceleration of the pole, setting in motion a feedback loop that results in the rapid dissipation of signal from $d\theta$. The resulting snap back of the cart towards the center causes the pole to swing up. As the pole falls and settles the same swing up behavior is repeated, and the controller is rewarded whenever the pole is upright.

As the search process continues, some of these controllers linger in the upright position longer than others, and by generation 128, the lingering duration is long enough for the pole to be kept balanced. Though this more complicated balancing mechanism is less reliable under variable weights than the swing-up and centering behaviors, the more reliable behaviors ensure that the system recovers and tries again until a balanced state is found. Notably, as these networks encode relationships and rely on tension between systems set against each other, their behavior is still consistent even with a wide range of shared weight values. For video demonstrations of the policies learned at various developmental phases of the weight agnostic topologies, please refer to the supplementary website.

WANN controllers for `BipedalWalker-v2` and `CarRacing-v0` (Figure 1, page 1) are likewise remarkable in their simplicity and modularity. The biped controller uses only 17 of the 25 possible inputs, ignoring many LIDAR sensors and knee speeds. The WANN architecture not only solves the task without training the individual weights, but uses only 210 connections, an order of magnitude fewer than commonly used topologies (2804 connections used in the SOTA baseline [34]).

The architecture which encodes stable driving behavior in the car racer is also striking in its simplicity (Figure 1, right). Only a sparsely connected two layer network and a single weight value is required to encode competent driving behavior. While the SOTA baseline [35] also gave the hidden states of a pre-trained RNN world model, in addition to the VAE's representation to its controller, our controller operates on the VAE's latent space alone. Nonetheless, it was able to develop a feed-forward controller that achieves a comparable score. Future work will explore removing the feed-forward constraint from the search to allow WANNs to develop recurrent connections with memory states.

The networks shows in Figure 1 (Page 1) were selected for both performance and readability. In many cases a great deal of complexity is added for only minimal gains in performance, in these cases we preferred to showcase more elegant networks. The final champion networks are shown in Figure 5.

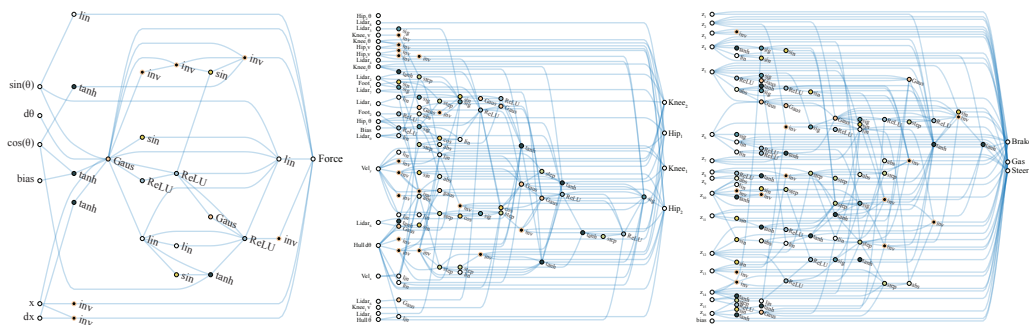

Figure 5: *Champion Networks for Continuous Control Tasks*
*Left to Right (Number of Connections)*: Swing up (52), Biped (210), Car Racing (245)
Shown in Figure 1 (Page 1) are high performing, but simpler networks, chosen for clarity. The three network architectures in this figure describe the champion networks whose results are reported.

**Classification** Promising results on reinforcement learning tasks lead us to consider how widely a WANN approach can be applied. WANNs which encode relationships between inputs are well suited to RL tasks: low-dimensional inputs coupled with internal states and environmental interaction allow discovery of reactive and adaptive controllers. Classification, however, is a far less fuzzy and forgiving problem. A problem where, unlike RL, design of architectures has long been a focus. As a proof of concept, we investigate how WANNs perform on the MNIST dataset [56], an image classification task which has been a focus of human-led architecture search for decades [14, 58, 96].

Even in this high-dimensional classification task WANNs perform remarkably well (Figure 6, Left). Restricted to a single weight value, WANNs are able to classify MNIST digits as well as a single layer neural network with thousands of weights trained by gradient descent. The architectures created still maintain the flexibility to allow weight training, allowing further improvements in accuracy.

| WANN | Test Accuracy |
|---|---|
| Random Weight | 82.0% ± 18.7% |
| Ensemble Weights | 91.6% |
| Tuned Weight | 91.9% |
| Trained Weights | 94.2% |

| ANN | Test Accuracy |
|---|---|
| Linear Regression | 91.6% [62] |
| Two-Layer CNN | 99.3% [15] |

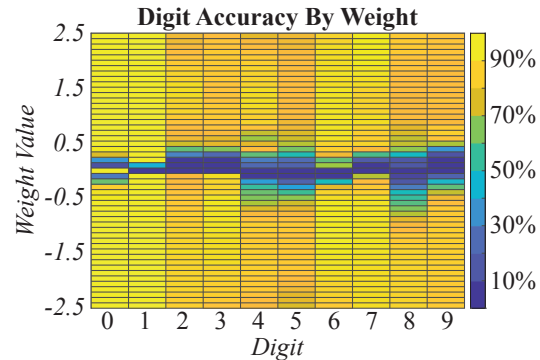

Figure 6: *Classification Accuracy on MNIST.*
*Left:* WANNs instantiated with multiple weight values acting as an ensemble perform far better than when weights are sampled at random, and as well as a linear classifier with thousands of weights.
*Right:* No single weight value has better accuracy on all digits. That WANNs can be instantiated as several *different* networks has intriguing possibilities for the creation of ensembles.

It is straight forward to sweep over the range of weights to find the value which performs best on the training set, but the structure of WANNs offers another intriguing possibility. At each weight value the prediction of a WANN is different. On MNIST this can be seen in the varied accuracy on each digit (Figure 6, Right). Each weight value of the network can be thought of as a distinct classifier, creating the possibility of using one WANN with multiple weight values as a self-contained ensemble.

In the simplest ensemble approach, a collection of networks are created by instantiating a WANN with a range of weight values. Each of these networks is given a single vote, and the ensemble classifies samples according to the category which received the most votes. This approach yields predictions far more accurate than randomly selected weight values, and only slightly worse than the best possible weight. That the result of this naive ensemble is successful is encouraging for experimenting with more sophisticated ensemble techniques when making predictions or searching for architectures.

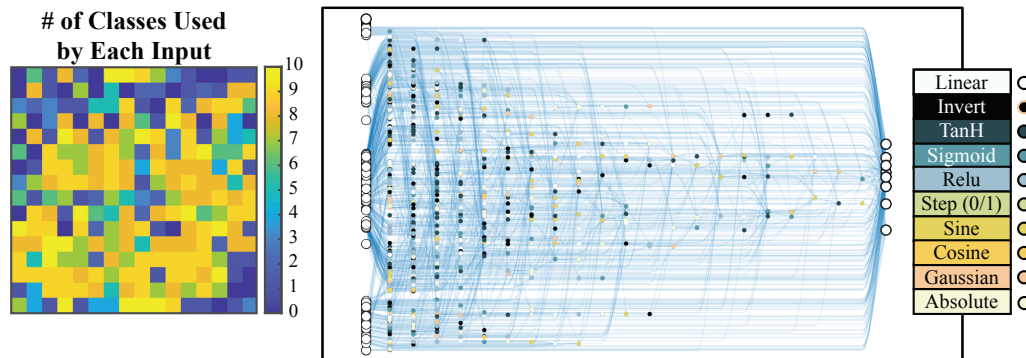

Figure 7: *MNIST classifier network (1849 connections)*
Not all neurons and connections are used to predict each digit. Starting from the output connection for a particular digit, we can trace the sub-network and also identify which part of the input image is used for classifying each digit. Please refer to the supplementary website for more detailed visualizations.

# 5 Discussion and Future Work

In this work we introduced a method to search for simple neural network architectures with strong inductive biases. Since networks are optimized to perform well using a shared weight over a range of values, this single parameter can easily be tuned to increase performance. Individual weights can be further tuned from a best shared weight. The ability to quickly fine-tune weights is useful in few-shot learning [21] and may find uses in continual learning [85] where agents continually acquire, fine-tune, and transfer skills throughout their lifespan, as in animals [120]. Inspired by the Baldwin effect [3], weight tolerant networks have long linked theories of evolution and learning in AI [1, 44, 102].

To develop a single WANN capable of encoding many different useful tasks in its environment, one might consider developing a WANN with a strong intrinsic bias for intrinsic motivation [84, 86, 97], and continuously optimize its architecture to perform well at pursuing novelty in an open-ended environment [62]. Such a WANN might encode, through a curiosity reward signal, a multitude of skills that can easily be fine-tuned for a particular downstream task in its environment later on.

While our approach learns network architectures of increasing complexity by adding connections, network pruning approaches find new architectures by their removal. It is also possible to learn a pruned network capable of performing additional tasks without learning weights [71]. A concurrent work [122] to ours learns a *supermask* where the sub-network pruned using this mask performs well at image recognition even with randomly initialized weights – it is interesting that their approach achieves a similar range of performance on MNIST compared to ours. While our search method is based on evolution, future work may extend the approach by incorporating recent ideas that formulate architecture search in a differentiable manner [67] to make the search more efficient.

The success of deep learning is attributed to our ability to train the weights of large neural networks that consist of well-designed building blocks on large datasets, using gradient descent. While much progress has been made, there are also limitations, as we are confined to the space of architectures that gradient descent is able to train. For instance, effectively training models that rely on discrete components [30, 48] or utilize adaptive computation mechanisms [29] with gradient-based methods remain a challenging research area. We hope this work will encourage further research that facilitates the discovery of new architectures that not only possess inductive biases for practical domains, but can also be trained with algorithms that may not require gradient computation.

That the networks found in this work do not match the performance of convolutional neural networks is not surprising. It would be an almost embarrassing achievement if they did. For decades CNN architectures have been refined by human scientists and engineers – but it was not the reshuffling of existing structures which originally unlocked the capabilities of CNNs. Convolutional layers were themselves once novel building blocks, building blocks with strong biases toward vision tasks, whose discovery and application have been instrumental in the incredible progress made in deep learning. The computational resources available to the research community have grown significantly since the time convolutional neural networks were discovered. If we are devoting such resources to automated discovery and hope to achieve more than incremental improvements in network architectures, we believe it is also worth trying to discover new building blocks, not just their arrangements.

Finally, we see similar ideas circulating in the neuroscience community. A recent neuroscience commentary, *"What artificial neural networks can learn from animal brains"* [120] provides a critique of how *learning* (and also *meta-learning*) is currently implemented in artificial neural networks. Zador [120] highlights the stark contrast with how biological learning happens in animals:

*"The first lesson from neuroscience is that much of animal behavior is innate, and does not arise from learning. Animal brains are not the blank slates, equipped with a general purpose learning algorithm ready to learn anything, as envisioned by some AI researchers; there is strong selection pressure for animals to restrict their learning to just what is needed for their survival."* [120]

This paper is strongly motivated towards these goals of blending innate behavior and learning, and we believe it is a step towards addressing the challenge posed by Zador. We hope this work will help bring neuroscience and machine learning communities closer together to tackle these challenges.

### Acknowledgments

We would like to thank our three reviewers for their helpful comments, and also express gratitude to Douglas Eck, Geoffrey Hinton, Anja Austermann, Jeff Dean, Luke Metz, Ben Poole, Jean-Baptiste Mouret, Michiel Adriaan Unico Bacchiani, Heiga Zen, and Alex Lamb for their thoughtful feedback.

## Footnotes

[2]We released a software toolkit not only to facilitate reproduction, but also to further research in this direction. Refer to the Supplementary Materials for more information about the code repository.

[3]Variations on these particular values had little effect, though weight values in the range $[-2, 2]$ showed the most variance in performance. Networks whose weight values were set to greater than 3 tended to perform similarly – presumably saturating many of the activation functions. Weight values near 0 were also omitted to reduce computation, as regardless of the topology little to no information is passed to the output.

[4]We conduct several independent search runs to measure variability of results in Supplementary Materials.

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
