[Supplementary Material]

# A   Supplementary Materials for Weight Agnostic Neural Networks

## A.1   Code Release

We release a general purpose tool, not only to facilitate reproduction, but also for further research in this direction. Our NumPy [9] implementation of NEAT [8] supports MPI [4] and OpenAI Gym [2] environments. All code used to run these experiments, in addition to the best networks found in each run, is referenced in the interactive article: `https://weightagnostic.github.io/`

## A.2   "Have you also thought about trying ... ?"

In this section, we highlight things that we have attempted, but did not explore in sufficient depth.

### A.2.1   Searching for network architecture using a single weight rather than range of weights.

We experimented with setting all weights to a single fixed value, e.g. 0.7, and saw that the search is faster and the end result better. However, if we then nudge that value by a small amount, to say 0.6, the network fails completely at the task. By training on a wide range of weight parameters, akin to training on uniform samples weight values, networks were able to perform outside of the training values. In fact, the best performing values were outside of this training set.

### A.2.2   Searching for network architecture using random weights for each connection.

This was the first thing we tried, and did not have much luck. We tried quite a few things to get this to work–at one point it seemed like we finally had it, poles were balanced and walkers walking, but it turned out to be a bug in the code! Instead of setting all of the weights to different random values we had set all of the weights to the *same* random value. It was in the course of trying to understand this result that we began to view and approach the problem through the lens of MDL and AIT.

### A.2.3   Adding noise to the single weight values.

We experimented adding Gaussian noise to the weight values so that each weight would be different, but vary around a set mean at each rollout. We only did limited experiments on swing-up and found no large difference, except with very high levels of noise where it performed poorly. Our intuition is that adding noise would make the final topologies even more robust to changes in weight value, but at the cost of making the evaluation of topologies more noisy (or more rollouts to mitigate the variance between trials). With no clear benefit we chose to keep the approach as conceptually simple as possible–but see this as a logical next step towards making the networks more weight tolerant.

### A.2.4   Using backpropagation to fine-tune weights of a WANN.

We explored the use of autograd packages such as JAX [3] to fine-tune individual weights of WANNs for the MNIST experiment. Performance improved, but ultimately we find that black-box optimization methods such as CMA-ES and population-based REINFORCE can find better solutions for the WANN architectures evolved for MNIST, suggesting that the various activations proposed by the WANN search algorithm may have produced optimization landscapes that are more difficult for gradient-based methods to traverse compared to standard ReLU-based deep network architectures.

### A.2.5   Why did you choose to use many different activation functions in the same network? Why not just ReLU? Wouldn't too many activations break biological plausibility?

Without concrete weight values to lean on, we instead relied on encoding relationships between inputs into the network. This could have been done with ReLUs or sigmoids, but including relationships such as symmetry and repetition allow for more compact networks.

We didn't do much experimentation, but our intuition is that the variety of activations is key. That is not to say that all of them are necessary, but we're not confident this could have been accomplished with only linear activations. As for biological corollaries, we're not going to claim that a cosine activation is an accurate model of a how neurons fire–but don't think a feed forward network of instantaneously communicating sigmoidal neurons would be any more biologically plausible.

## A.3  MNIST

The MNIST version used in this paper is a downsampled version, reducing the digits from [28x28] to [16x16], and deskewed using the OpenCV library[1]. The best MNIST network weight was chosen as the network with the highest accuracy on the training set.

To fit into our existing approach MNIST classification is reframed as a reinforcement learning problem. Each sample in MNIST is downsampled to a 16x16 image, deskewed, and pixel intensity normalized between 0 and 1. WANNs are created with input for each of the 256 pixels and one output for each of the 10 digits. At each evaluation networks are fed 1000 samples randomly selected from the training set, and given reward based on the softmax cross entropy. Networks are tested with a variety of shared weight values, maximizing performance over all weights while minimizing the number of connections.

## A.4  Hyperparameters and Setup

All experiments but those on Car Racing were performed used 96 core machines on the Google Cloud Platform. As evaluation of the population is embarrassingly parallel, populations were sized as multiples of 96 to make efficient use of all processors. Car Racing was performed on a 64 core machine and the population size used reflects this. The code and setup of the VAE for the Car Racing task is taken from [7], were a VAE with a latent size of 16 was trained following the same procedure as [7]. Tournament sizes were scaled in line with the population size. The number of generations were determined after initial experiments to ensure that a majority of runs would converge.

|  | SwingUp | Biped | CarRace | MNIST |
|---|---|---|---|---|
| Population Size | 192 | 480 | 64 | 960 |
| Generations | 1024 | 2048 | 1024 | 4096 |
| Change Activation Probability (%) | 50 | 50 | 50 | 50 |
| Add Node Probability (%) | 25 | 25 | 25 | 25 |
| Add Connection Probability (%) | 25 | 25 | 25 | 25 |
| Initial Active Connections (%) | 50 | 25 | 50 | 5 |
| Tournament Size | 8 | 16 | 8 | 32 |

## A.5  Results over multiple independent search runs

For each task a WANN search was run 9 times. At regular intervals the network in the population with the best mean performance was compared to that with the previously best found network. If the newer network had a higher mean, the network was evaluated 96 or 64 times (depending on the number of processors on the machine), and if the mean of *those* evaluations was better than the previous best network, it was kept as the new 'best' network. These best networks were kept only for record keeping and did not otherwise interact with the population.

These best networks at the end of each run were reevaluated thirty times on each weight in the series $[-2, -1.5, -1, -0.5, 0.5, 1, 1.5, 2]$ and the network with the best mean chosen as the champion for more intensive analysis and examination. Shown below are the results of these initial tests, both as individual runs and as distributions of performance over weights.

Figure 1: *Swing-up Performance over Multiple Runs.*
*Left*: Performance per weight value of best network found in each of 9 runs.
*Right*: Average performance of best networks found at end of each of 9 runs. Performance is shown by top weight, top quartile of weights, top half of weights, and over all weights.

Figure 2: *Biped Performance over Multiple Runs.*
*Left*: Performance per weight value of best network found in each of 9 runs.
*Right*: Average performance of best networks found at end of each of 9 runs. Performance is shown by top weight, top quartile of weights, top half of weights, and over all weights.

Figure 3: *Car Racing Performance over Multiple Runs.*
*Left*: Performance per weight value of best network found in each of 9 runs.
*Right*: Average performance of best networks found at end of each of 9 runs. Performance is shown by top weight, top quartile of weights, top half of weights, and over all weights.

### A.6   Optimizing for individual weight parameters

In our experiments, we also fine-tuned individual weight parameters for the champion networks found to measure the performance impact of further training. For this, we used population-based REINFORCE, as in Section 6 of [10]. Our specific approach is based on the open source `estool` [5] implementation of population-based REINFORCE. We use a population size of 384, and each agent performs the task 16 times with different initial random seeds for Swing Up Cartpole and Bipedal Walker. The agent's reward signal used by the policy gradient method is the average reward of the 16 rollouts. For Car Racing, due to the extra computation time required, we instead use a population size of 64 and the average cumulative reward of 4 rollouts to calculate the reward signal. All models trained for 3000 generations. All other parameters are set to the default settings of `estool` [5].

For MNIST, we use the negative of the cross entropy loss as the reward signal, and optimize directly on the training set with population-based REINFORCE.

### A.7   Fixed Topology Baselines

For Bipedal Walker, we used the model and architecture available from `estool` [5] as our baseline. To our knowledge, this baseline currently, at the time of writing, achieves the state-of-the-art average cumulative score (over 100 random rollouts) on Bipedal Walker as reported in [6].

In the Swing Up Cartpole task, we trained a baseline controller with 1 hidden layer of 10 units (71 weight parameters), using the same training methodology as the one used to produce SOTA results for the Bipedal Walker task mentioned earlier. We experimented with a larger number of nodes in the hidden layer, and an extra hidden layer, but did not see meaningful differences in performance.

For the Car Racing baseline, we used the code and model provided in [7] and treated the 867 parameters of the controller as free weight parameters, while keeping the parameters of the pre-trained VAE and RNN fixed. As of writing, the average cumulative score (over 100 random rollouts) produced by [7] for Car Racing is currently the state-of-the-art. As mentioned in the main text, for simplicity, the WANN controller has access only to the pre-trained VAE, and not to the RNN.