[Reviews · NeurIPS 2019]

Reviewer 1



Originality: This is the first approach that I know of which attempts to train on the network architecture independently of the weight values. Significance: The relationship to neuroscience is somewhat remote, synapses in the brain do have different connection strengths. The significance of the empirical results is rather low as their resulting network architecture looks rather shallow and the task involved seem to be solvable with shallow networks trained other evolutionary-like algorithms. Clarity: The language is quite good. However, many technical details are not spelled out. I think the description of the algorithm l. 123 to 129 is not sufficient to understand the algorithm. Quality: I think that it is unfortunate that the authors do not analyze in more details their training algorithm in comparison to other approaches, or variants of their own algorithm. -------- Second review ----- I am glad that the positive reviews of the other reviewers got me to look again at the paper. After another reading, I do find that the originality of this work should be valued a lot more than I did it in my first review. It is still hard for to me know if the performance reported is trivial or satisfactory as a proof of concept. Despite this, I change the "overall score" of my review. One big improvement is that the authors promise to add a pseudo code in the supplements I hope that the details of the algorithm will make it more accessible. I also think it would be great to discuss somewhere what is new in this algorithm in comparison with the NEAT framework.

Reviewer 2



Originality: This paper draws from many fields (especially neural architecture search), but its core is a unique and powerfuly original idea. Quality: The execution of this idea by the authors is thorough and the results are compelling. The scholarship evident in this work is also exemplary. Clarity: This paper is extremely well-written and easy to follow. One small exception, however, was how the authors discussed their use of a single shared weight across all the connections in the network. At the beginning of the paper their stated ambition to use a single weight was confusing: why limit the network so much? In section 3 they explain this choice clearly and convincingly: the dimensionality of the full weight space is too large to effectively explore, so they test only a 1-D manifold within that space. But after reading the continuous control results section it actually seemed like even this 1-D weight manifold was over-parameterized: for nearly every problem & network the performance was either good with large absolute shared weight regardless of sign, or good with one or the other sign of large shared weight (e.g. the bipedal walker network that only performs well with large negative weights). This suggests that the same results could be obtained without randomizing the shared weight, but just using a single value throughout training (e.g. 1). Besides the ensemble classifier discussed in the last section, is there a strong advantage to using a shared random weight rather than a shared fixed weight? Does it make the networks more easily fine-tuned when the individual weights are trained? It could help improve the paper somewhat if the authors clarified the importance and reasoning behind using a single shared weight early on (i.e. in the intro). Significance: This work is highly significant: it introduces a new realm of problems (weight-agnostic neural networks) that have strong analogues in biological systems. This work opens many new avenues for research and is sure to be widely cited.

Reviewer 3



Review update: Thanks for your response. I have read it and I still like the paper. Nice work! Summary The paper uses architecture search based on genetic algorithms to find architectures on simple reinforcement learning tasks and MNIST that only have a single shared weight parameters. They demonstrate that these networks perform competitively with hand-picked and trained architectures. Strengths and weaknesses The paper contributes to the recent body of papers on architecture search and explores the inductive bias induced by the network architecture in a very extreme case where the network only has a single parameter. Given that most networks currently use standardized architectural components and heavily focus on weight training, this paper offers a refreshing view on different aspects responsible for a good inductive bias. In particular, it's in line with a recent report (https://arxiv.org/abs/1904.01569) that demonstrates that the graph architecture class (i.e. random connect vs. small word etc.) seems to have a relatively larger influence on the performance than the weights of the network. Apart from a few minor comments the paper is clearly written, and represents a significant and original contribution to this line of research. I think the paper has two main weaknesses: 1. The strategy is obviously only applicable to very small problems/networks. A demonstration that networks composed of repeating/recursive simple network motifs can solve next level complex problem such as CIFAR, would strengthen the paper. 2. Apart from the demonstration that a single weight parameter is enough, the paper offers little insight into the quantification of inductive biases by architecture. For instance, how many networks perform well? Do well performing networks differ strongly (in their components and the class of functions they implement)? Minor comments - Is there a reason why you don't use a delete operation? - Table 1 doesn't say what the number is. I guess reward, but it would be good to say it explicitly. - A bit of information on how the VAE for MNIST was trained would make the paper more self-contained.

[Author Response · NeurIPS 2019]

We thank all our reviewers for their feedback! We will respond to (R2, R3) separately to R1 due to different concerns.

**R2 and R3:** We thank R2 and R3 for their vote of confidence and giving this work at high score of 9 and 8 respectively. It means a lot to us – to see our ideas accepted by our peers at NeurIPS who also believe that our "work opens many new avenues for research and is sure to be widely cited" (R2). We agree with R2 that the importance behind random single weight choice, rather than single weight should be explained clearly in the intro, and we will update the draft to do so.

We experimented with setting all weights to a single fixed value, e.g. 0.7, and saw that the search is faster and the end result better. However, if we then nudge that value by a small amount, to say 0.6, the network fails completely at the task. By training on a wide range of weight parameters, akin to training on uniform samples weight values, networks were able to perform outside of the training values. In fact, the best performing values were outside of this training set.

The work that R2 pointed out, Zador2019, is indeed an inspiring critique of deep learning from the neuroscience community, and our work shares similar motivations. We will cite and discuss this work in our revised paper. As an aside, the *Animal AI Olympics* contest and related *Learning Transferable Skills* workshop, from same organizers, at NeurIPS2019 will discuss similar themes and we are excited to see more ideas in this direction from both communities.

We agree with R3 that scaling up is the next step. Currently we are exploring hybrid approaches (in the direction of *arxiv:1904.01569* R3 mentioned that employs 3x3 convs and random graphs) and indirect encodings (i.e. HyperNEAT, Stanley2009) to scale WANNs architectures to scales able to compete on benchmarks such as ImageNET and Atari. We wish to take the time to conduct this investigation thoroughly, and plan to report the findings in a follow up paper. Qualitative analysis of sample networks was done in Section 4, and in future work we plan to develop a method of automated quantitative analysis, as suggested by R3, to identify promising structures, modules, and motifs. The 'delete' operation suggested by R3 would have interesting effects but in the end we tried to present the simplest algorithm we thought could still work – though have no reason to think that a delete operation couldn't make even more compact WANNs. We would also like to thank R3 for the other minor suggestions, we will clarify the labels and information.

**R1:** We thank R1 for their critical review – that the contribution of this work is not well-defined by performance metrics makes it all the more important that its value is clearly expressed. We believe it is important for us to communicate and emphasize the merits of this work beyond benchmark scores.

The point of the paper was *not* to propose a new RL algorithm which meets or beats SOTA results, but to explore the importance of biases in neural network architectures. Our goal is to provide an existence proof that neural networks can be automatically designed to encode such biases. The motivation was not to outperform weight training algorithms but to explore an orthogonal approach – "how well can we perform *without* weight training?" We will emphasize the exploratory nature of the paper more heavily, in the hopes that it will be read in this frame of mind.

In the spirit of this extreme experiment the algorithm used was purposefully kept simple. It is not a lack of understanding from R1 that the technical contribution of the EA is minimal – this was precisely the intention. The innovation introduced in our method is to evaluate networks with sampled shared weight values rather than optimizing the weights, an approach we have not encountered in previous work. In the interest of clarity we will add pseudocode of the procedure used to search for WANNs in the appendix and emphasize in the text that we are presenting a purposefully minimal method.

The car racing experiment was conducted to show how WANNs can work with existing deep learning approaches, to help bridge different fields. As for concerns raised about the effectiveness of the network architecture discovered, we demonstrate comparable performance to [32] that used both a VAE and RNN, while in our work we used the VAE alone. In [32], it was shown that a VAE-only approach does not perform well with a 1 or 2 layer controller, where we have shown that the WANN was effective for this particular task using VAE-only.

Our original intention was to focus only on continuous-control RL experiments, and decided to run MNIST "for fun" near the end of the project. We could have confined the paper to only RL experiments (most RL papers don't run MNIST experiments), but chose to report the MNIST results to highlight both the benefits and limitations of the approach. We also think that a result of 80-90% (whether good or bad) with a randomly initialized network is interesting, especially compared to chance accuracy. Even with the MNIST section omitted, we believe the paper with only RL experiments still warrants a score above 3, a clear rejection score, as the contributions are valuable to the NeurIPS community.

Finally, we do believe there is a connection to the neuroscience field. In addition to the literature already cited in the paper, we see similar ideas circulating in the neuroscience community: R2 pointed out a recent neuroscience paper, "What Artificial Neural Networks can Learn from Animal Brains" (Zador2019) whose central theme is that "The first lesson from neuroscience is that much of animal behavior is innate, and does not arise from learning. Animal brains are not the blank slates, equipped with a general purpose learning algorithm ready to learn anything, as envisioned by some AI researchers; there is strong selection pressure for animals to restrict their learning to just what is needed for their survival." Our work is strongly motivated towards these goals of blending innate behavior and learning, and believe it will help bring neuroscience and machine learning communities closer together to tackle these challenges.

[Meta-Review · NeurIPS 2019]

This paper examines the power of network architecture in isolation, without any contribution from synaptic weight training, to solve ML tasks. Specifically, the paper examines the extent to which neural networks with random weights can perform tasks if the architecture has been optimized appropriately. The authors provide a novel algorithm for conducting this optimization on architectures, and show that they can achieve surprisingly good results with random weights (e.g. ~80% on MNIST). This paper demonstrates the potential power of architecture optimization procedures, and provides a method for architecture optimization that may be very useful. The results may also be more broadly interesting with respect to questions of seeking appropriate inductive biases in ML. It will be of significant interest to the NeurIPS community.